# The Relationship between Physical Activity and Mental Depression in Older Adults during the Prevention and Control of COVID-19: A Mixed Model with Mediating and Moderating Effects

**DOI:** 10.3390/ijerph20043225

**Published:** 2023-02-12

**Authors:** Guoyan Xiong, Caixia Wang, Xiujie Ma

**Affiliations:** 1School of Wushu, Chengdu Sport University, Chengdu 610041, China; 2School of Physical Education, Handan University, Handan 056005, China; 3Chinese Guoshu Academy, Chengdu Sport University, Chengdu 610041, China

**Keywords:** COVID-19, physical activity, mental depression, social support, self-efficacy, older adults

## Abstract

Background: Several studies have found a strong relationship between physical activity and mental depression in older adults. Despite this, the social isolation, limited physical activity, and decreased social interactions caused by the 2020 COVID-19 pandemic control measures of “home isolation and reduction unnecessary travel” had a significant mental impact on older adults. Objective: the goal of this study was to look into the complex effects of physical activity participation on mental health in older adults during COVID-19 prevention and control and the relationship between physical activity and mental depression in older adults through the mediating effect of self-efficacy and the moderating effect of social support. Methods: The Physical Activity Rating Scale (PARS-3), the Center for Streaming Depression Scale (CES-D), the Self-Efficacy Scale (GSES), and the Social Support Rating Scale (SSRS) were used to assess 974 older adults in five urban areas of Chengdu, China. The SPSS was used to analyze the collected data using mathematical statistics, linear regression analysis, and the AMOS to construct the research model. Results: The study’s findings revealed that self-efficacy mediated the relationship between physical activity and mental depression in older adults (*β* = −0.101, 95%CI (−0.149, −0.058)), and social support moderated the relationship between physical activity and mental depression in older adults (t = −9.144, *p* < 0.01). Conclusions: Physical activity reduces psychological depressive symptoms in older adults and modulates psychological depression in older adults via the mediation efficacy of self-efficacy and the moderating effect of social support.

## 1. Introduction

The coronavirus disease 2019 (COVID-19) has become a public health emergency of international concern since the associated outbreak [1]. Its high contagiousness and rapid spread not only endanger public health but also have a significant impact on production, lifestyle, and quality of life. Moreover, in many cases, it has caused unbearable psychological stress. The public has faced severe physical and mental health challenges, particularly vulnerable groups such as the elderly. It has been demonstrated that older adults are more vulnerable to physical and psychological effects during COVID-19 [2] and that psychological changes are more pronounced in those over 60 years old than in other age groups [3]. The most common mental problems among older adults during COVID-19 outbreak prevention and control are panic disorder, anxiety disorder, and depression [4], and the negative impact of depression on older adults cannot be overlooked. Depression is typically characterized by a depressed mood, pessimism, appetite disorders or sleep deprivation, low self-worth, slowed thinking, and reduced responsiveness. These issues frequently become chronic, resulting in disability or, in the worst-case scenario, suicide attempts [5].

A growing number of studies has recently discovered that physical activity has been shown to be as effective as antidepressant medication in alleviating depression [6]. Furthermore, engaging in moderate physical exercise improves older people’s mental health [7], and physical activity not only boosts good feelings [8], but it also improves cognitive and emotional functioning, which has a favorable impact on physical functioning, mental health status, quality of life, and subjective well-being [9,10].

There have been numerous studies on the relationship between physical activity and mental health in older adults during COVID-19, but few have examined the intricate mechanisms underlying this relationship. In addition, existing research studies still have blind spots, such as the part that self-efficacy plays in the association between physical activity and psychological depression in older adults. Current studies on self-efficacy and mental depression during the pandemic have included children [11], adolescents [12], healthcare workers [13], and pregnant women [14], but no studies on the effect of self-efficacy on mental depression in older adults during COVID-19 pandemic prevention and control have been found. Moreover, there is still space for advancements in methodology since earlier studies only employed a single statistical method and had tiny sample sizes.

In the context of a rapidly aging population and the COVID-19 pandemic, older adults’ physical and mental health status as well as physical activity levels are deteriorating year by year. Exploring the impact of physical activity on depression in older adults during the COVID-19 pandemic prevention and control could help them achieve healthy aging, strengthen their immunity, and reduce the national medical burden, among other things. Therefore, this study chose self-efficacy as a mediating variable and social support as a moderating variable to investigate the relationship between physical activity and mental depression among older adults during pandemic prevention and control. This study focused on the mechanisms and effects of physical activity, self-efficacy, social support, and mental depression.

### 1.1. Physical Activity and Mental Depression

Prior to the COVID-19 outbreak, mental health problems in older adults were already common. According to a survey in 2017, 10–15% of older adults had clinically significant depression [15]. The COVID-19 outbreak exacerbated the mental health crisis in the elderly population. During the outbreak prevention and control period, a series of measures such as “home isolation and reduction of unnecessary travel” reduced not only physical activity and social interaction among older adults but also sleep quality, which together may have resulted in a high prevalence of symptoms such as frailty and depression [16]. Many researchers have found a strong relationship between physical activity and mental depression in older adults [17,18]. Physical activity, for example, has been shown to be a protective factor against depression [19], and long-term physical activity is an effective and convenient way to prevent depressive symptoms in older adults [20]. Physical activity has been suggested to have a therapeutic effect on depression [21,22]. For example, physical exercise is beneficial in the treatment of depression’s “negative effects”, such as loss of interest, reduced activity, and mood instability [23], and various types of physical activity may also be an effective non-pharmacological treatment for depression in older adults [24]. In addition, regular physical activity not only relieves psychological stress and anxiety disorders in older adults but also helps in the maintenance and development of physical and mental health [25]. As a result, physical activity among older adults during COVID-19 prevention and control may help to alleviate their psychological depression. The following hypothesis is proposed in this study based on the above analysis:

**H1:** 
*Physical activity has a significant negative effect on psychological depression.*


### 1.2. Mediation of Self-Efficacy

As psychology research has progressed, the benefits of physical activity on mental health have gained widespread recognition [26]. Physical activity has a significant impact on self-efficacy according to the majority of academics. Self-efficacy and physical activity were found to have a positive correlation in college students, teenagers, and older adults, with the relationship becoming stronger the more physical activity they engaged in [27,28,29]. Furthermore, those who exercised regularly had much higher self-efficacy than those who were sedentary, and regular exercise was shown to have a positive impact on one’s mental health in addition to being the best way to raise self-efficacy [30]. The development and enhancement of self-efficacy have been shown to reduce depressive symptoms [31]. Self-efficacy is recognized as an important factor in determining health [32] and can be used to predict depressive symptoms [33]. Depression may result from low self-efficacy [34], whereas high self-efficacy is associated with a significantly lower risk of depression [35]. Despite previous studies showing the benefits of physical activity to promote self-efficacy, could self-efficacy indirectly affect psychological depression in older adults when physical activity acts upon depression? As a result, the following theory was explored in this study:

**H2:** 
*Self-efficacy acts as a mediator in the connection between physical activity and mental depression.*


### 1.3. Moderation of Social Support

A growing body of research has shown that social support has a general beneficial effect on the development of psychological well-being in individuals [36] and that higher levels of social support can reduce the negative effects of stressful events on the individual’s mind and body, reduce the bonding relationship between stress and depression, and thus reduce the degree and generation of depression [37]. Studies on social support and older adults found that social support allows older adults to feel cared for and loved, which is beneficial for the maintenance and development of physical and mental health, and that family and friend support reduces psychological stress and improves well-being, promoting psychological health and preventing depression in later life [38]. Furthermore, there is mounting evidence showing that social support is an important factor in influencing behavioral change and that the more social support people receive, the more physically active they are [39], an important factor in influencing physical activity levels [40]. Physical activity has a greater impact on individual psychological depression when there is a high level of social support. In other words, social support can help to strengthen the link between physical activity and self-efficacy or depression. As a result, the following hypothesis was proposed:

**H3:** 
*Social support moderates the relationship between physical activity and mental depression.*


## 2. Materials and Methods

### 2.1. Hypotheses and Conceptual Model

This study developed a research model of the mediating and moderating mechanisms of physical activity and mental depression in older adults during pandemic prevention and control, using mental depression as the dependent variable, physical activity as the independent variable, self-efficacy as the mediating variable, and social support as the moderating variable. This model was based on the aforementioned literature review and the proposed hypotheses (Figure 1). The relationships between mental depression, physical activity, self-efficacy, and social support were revealed by the model. First, there was a direct link between physical activity and mental depression; second, physical activity was indirectly linked to mental sadness through self-efficacy; and third, social support might step in to strengthen the link between physical activity and mental depression.

### 2.2. Participants and Procedures

This study examined the physical activity and mental depression of older individuals in five Chengdu urban regions: Qingyang, Jinniu, Chenghua, Jinjiang, and Wuhou, using random sampling. The number of surveys in each urban region was restricted to 200 in order to ensure proper sample dispersion. In the end, 1257 questionnaires—1012 offline and 245 online—were retrieved, yielding a total of 974 valid questionnaires and a valid recovery rate of 77.49%. Figure 2 illustrates the screening process of the study sample. Inclusion and exclusion criteria were established because the purpose of this study was to investigate the connection between physical activity and mental depression in older people. The inclusion standards were as follows: (1) people over the age of 60; (2) permanent residents with Chengdu household registration; (3) experience with the new coronavirus being inhibited and controlled; (4) informed consent and voluntary participation; and (5) unrestricted communication. The exclusion criteria were as follows: (1) a questionnaire completion time of less than 90 s and (2) repeated and invalid answers.

A simultaneous on-site distribution of paper surveys and online questionnaires, which took 3 to 5 min to complete, were used in the research process. Two to three researchers who looked for places where older people might engage in physical activity distributed the questions both locally and online using Questionnaire Star to the target audience. The Chengdu Institute of Physical Education gave its prior ethical approval. The locations in the five urban areas that had been subject to closure control management were identified using the official microblogging platform of Healthy Chengdu before the questionnaires were distributed, and the questionnaires were then given out to the designated groups of people at the designated locations. The study’s initial purpose, the use of the study’s data, confidentiality of personal data, and its risks were all described to the participants by the researchers before they began the questionnaire. Thereafter, the subjects were presented with and asked to sign an informed consent form. When necessary, researchers were required to explain questions to participants that they did not understand and to provide oral interpretation in the participants’ dialects. Participants were given a red packet or gift as a token of appreciation after completing the survey.

### 2.3. Control Variables

Since the level of psychological depression affects older adults’ subjective well-being and is influenced by individual and social factors, gender, age, education, income, and health status were used as control variables to reduce the risk of statistical bias.

### 2.4. Measurements

#### 2.4.1. Physical Activity

The Physical Activity Rating Scale (PARS-3) revised by Liang, D.C. et al. was used to assess physical activity in the elderly. The amount of exercise was examined in terms of three aspects, namely intensity, time, and frequency of participation in physical activity, using a 5-point scale [41]. The intensity is divided into four categories: light exercise (such as walking, doing radio gymnastics, etc.), less-intense exercise (such as recreational ping pong, jogging, tai chi, etc.); more intense and lasting exercise of medium intensity (such as cycling, running, etc.); heavy intensity with shortness of breath and much sweating but not lasting exercise (such as playing badminton, basketball, tennis, etc.); and heavy and lasting exercise with shortness of breath and much sweating (such as racing, sets of aerobics exercises, swimming, etc.); time span is 10–60 min; frequency is calculated as the number of physical activities performed per month and per week and contains less than 1 time a month, 3 to 5 times a week, 2 to 3 times a month, approximately 1 time a day, and 1 to 2 times a week. Specifically, amount of exercise = intensity × time × frequency. Intensity and frequency were scored from 1 to 5 levels and recorded as 1 to 5 points; time was scored from 1 to 5 levels recorded as 0 to 4 points, with a maximum score of 100 points and a minimum score of 0 points. The activity level assessment criteria are as follows: 19 for low exercise levels, 20–42 for moderate exercise levels, and 43 for high exercise levels. In this study, the Cronbach coefficient for this scale was 0.797.

#### 2.4.2. Mental Depression

Jie Zhang et al. revised the Streaming Center Depression Scale (CES-D) to assess depression. It has 16 negative affective items and 4 positive affective items for a total of 20 items, 4 of which are reverse scoring questions [42]. To complete the form, subjects were asked to rate the frequency of symptoms in the previous week on a 4-point scale, with 0 indicating rarely or never (less than 1 day), 1 indicating occasionally (1–2 days), 2 indicating occasionally or half the time (3–4 days), and 3 indicating most of the time or continuously (5–7 days), with a total score of 0–60 points. The scale was divided into scoring criteria: 16 points for no depression, 16 total points for suspected depression, and 20 points for some degree of depression. In this study, the Cronbach coefficient of the scale was 0.952.

#### 2.4.3. Self-Efficacy

In this study, the General Self-Efficacy Scale (GSES) developed by Schwarzer et al. was used, which is a unidimensional scale with 10 items [43]. A 4-point Likert scale was used, with a score of 1 indicating “not at all correct”, 2 indicating “somewhat correct”, and 3 indicating “mostly correct.” The higher the score, the greater the self-efficacy. The overall score ranged from 10 to 40, with 10–20 for low self-efficacy, 21–30 for moderate self-efficacy, and 31–40 for high self-efficacy. In this study, the Cronbach coefficient for this scale was 0.934.

#### 2.4.4. Social Support

The Social Support Rating Scale (SSRS) was used in this study. It was developed by Xiao Shuihui and other mental health workers on the basis of foreign scales and has 10 entries. It includes three dimensions of objective support, subjective support, and utilization of social support [44]. Articles 1–5 and 8–10 were scored on a 4-point scale, with a value of 1–4 points; articles 6–7 were scored on a 2-point scale, with a value of 0–1 points, and the answer “no source” received 0 points. The scale’s total score ranged from 12 to 66, with 0–22 indicating low social support, 23–44 indicating moderate social support, and 45–66 indicating high social support. Higher total scores and scores on each scoring scale indicate greater social support. In this study, the Cronbach coefficient for this scale was 0.947.

### 2.5. Statistical Analysis

First, SPSS 25.0 software (IBM, Armonk, NY, USA)was used to analyze the valid survey data. The model and the scale’s structural validity were assessed using the Amos 24.0 software package(SPSS, Chicago, USA). Validated factor analysis (CFA) was used in this study to test for convergent validity, the factor loading coefficient values showed the correlations between the factors and the analyzed items, and the combined reliability and validity of the scales were further examined using the average variance extracted values (AVE) and the factor loading combined reliability (CR).

Second, Pearson correlation coefficients were used to assess the linear relationships between physical activity, self-efficacy, and psychological depression in older persons. The study then used the AMOS 24.0 software program to conduct a goodness-of-fit analysis of the conceptual construct mediation model to confirm the mediating function of self-efficacy between physical activity and psychological depression, and the bootstrap method was used to assess whether there is a mediating effect of self-efficacy between physical activity and mental depression in older adults. The bootstrap approach is commonly used to test the mediating impact. Repeated sampling from the original sample is used in this technique to determine the coefficient of the mediating effect’s significance using a 95% confidence interval [45].

Finally, linear regression was used to test the role of social support as a moderator in the relationship between physical activity and mental depression in older adults. The three-step test of the hierarchical moderated regression (HMR) analysis was used in this study to test the moderating effects, and the interaction term of the variables was used to test the moderating effects. To elaborate, the empirical test was carried out using the following procedures, and SPSS 25.0 was utilized to conduct the statistical analysis of physical activity. Correlation analysis was used to perform the initial test of hypothesis testing based on the test of common method bias, and then, linear regression analysis was used to perform the moderating model test. Step-by-step analyses were conducted using three linear regression models: first, model 1 (M1) was fitted with gender, age, education, income, and health condition as control variables and different physical activity scores as independent variables for regression; model 2 then added moderating variables (social support) to model 1; and finally, model 3 added interaction terms (product terms of independent and moderating variables) to model 2.

This study divided the mean of social support and physical activity (M) plus or minus one standard deviation (SD) into high and low subgroups and plotted a simple slope test of social support between physical activity and mental depression in order to clearly present the moderating effect of social support (Figure 3) based on the method suggested by Aiken and West [46].

## 3. Results

### 3.1. Analysis of the Sample Situation

A total of 974 valid questionnaires were recovered from this survey, as shown in Table 1. Males made up 490 of them (50.3%), while females made up 484 (49.7%). The majority of them, i.e., 303 of them, were aged 60 to 64 (31.1%); a higher percentage of them had completed primary school (26.8%); the majority of the older individuals had an annual income of between USD 2000 and 3000 (30%); and a higher percentage of the elderly were in very good health (45%).

During the new pandemic, high-risk areas implemented control measures such as “foot out of the district, door-to-door service”, and no new infections for 7 consecutive days result downgrading to a medium-risk area, and no new infections for 3 consecutive days result in downgrading to a low-risk area, according to information released by the Chengdu Municipal Health and Wellness Commission. A medium-risk area can be lowered to a low-risk area by implementing “foot out of the district, staggered pick up” and other control measures and no new infections for seven consecutive days. Low-risk regions take preventive steps such as “personal protection and avoid gathering” and allow people to travel freely under the circumstances [47]. As a result, a high-risk area needs at least 10 days to be released from control, while a low-risk area takes at least 7 days. It was found through the survey that more older adults in the respondent group engaged in low levels of physical activity (43.3%), 52.8% of the group were found to have signs of depression, 38.4% of respondents had low levels of self-efficacy, and 65.2% of respondents reported moderate levels of social support. Although the sample data cannot be used to directly infer a relationship between physical activity and psychological depression in older adults, it does allow readers and other researchers to fully comprehend the sample’s characteristics.

### 3.2. Reliability and Validity Tests

First, as shown in Table 2, all question items had factor loadings that ranged from 0.664 to 0.910, all of which were higher than 0.6, demonstrating a significant association between the factors and the analyzed terms. It was discovered that each factor’s AVE and CR values were greater than 0.5 and greater than 0.7, respectively, indicating that the data had good construct validity and consistency. In addition, it is clear to see from the model fit indicators of the validation factor analysis results in Table 3 that the model fits well: χ^2^/*df* < 3, GFI > 0.9, AGFI > 0.9, CFI > 0.9, NFI > 0.9, NFI > 0.9, RMSEA < 0.05. The questionnaire survey in this research, therefore, has a high level of internal validity and reliability.

### 3.3. Descriptive Statistics and Correlation Analysis

Table 4 displays descriptive statistics. According to Table 4, physical activity had a significant negative correlation with mental depression (r = −0.649, *p* < 0.01). That is, the likelihood of experiencing depression symptoms decreases with increasing physical exercise. There was also a significant positive correlation with self-efficacy (r = 0.551, *p* < 0.01). That is, physical activity and self-efficacy levels both grew in the same manner. Moreover, self-efficacy had a significant negative correlation with mental depression (r = −0.485, *p* < 0.01). In other words, the risk of having a depression mood decreases with a higher sense of self-efficacy.

Furthermore, with correlation coefficients of 0.233 and −0.081, respectively, social support had strong positive and negative correlations with physical activity and mental depression, both at a significant level of 0.01 or less. That is, there is an inverse or diminishing growth between social support and psychological depression, whereas there is a homogeneous increase between social support and physical activity. In this paper, hypotheses H1, H2, and H3 were initially supported.

### 3.4. Mediating Analysis

The standardized path coefficient model of physical activity and self-efficacy affecting mental depression is shown in Figure 4. According to Table 5’s model fit indices for the mediating role of physical activity, self-efficacy, and mental depression, the model of how physical activity affects mental depression fits well: χ^2^/df < 3, GFI > 0.9, AGFI > 0.9, CFI > 0.9, NFI > 0.9, NFI > 0.9, and RMSEA < 0.05. As shown in Figure 4 the path coefficient of physical activity → mental depression was significant (*β* = −0.55), indicating a direct effect of physical activity on mental depression. Thus, hypothesis H1 holds. The path coefficients for both physical activity → self-efficacy and self-efficacy (*β* = 0.55) → mental depression (*β* = −0.18) were significant, suggesting a mediating effect of self-efficacy between physical activity and mental depression. Thus, hypothesis H2 holds.

A minimum of 1000 replicate samples of the original sample are required for the bootstrap-mediated effects test [48]. If the bootstrap-mediated effects test results show that the bootstrap test CI does not contain a value of 0, the indirect effect is present [49]. As a result, the mediation effect test was carried out in this study by estimating the bootstrap 95% CI of the mediation effect by sampling 2000 times, and the results are shown in Table 6. The standard error (S.E.) of the indirect effect of physical activity → self-efficacy → mental depression was 0.023 with a Z value of −4.391, and the indirect effect bootstrap 95% CI generated by this path did not contain a 0 value, indicating a significant mediating effect of self-efficacy between physical activity and mental depression. The standard error of the direct effect of physical activity → mental depression was 0.046, the Z value was −11.913, and the bootstrap 95% CI for the indirect effect of this pathway did not contain a 0 value. The standard error of the total effect of physical activity → mental depression was 0.034, the Z value was −19.088, and the bootstrap 95% CI for the total effect of this pathway did not contain a 0 value. This indicates that physical activity had a significant effect on both the direct and total effects of mental depression.

### 3.5. Moderating Analysis

The goal of model 1 was to investigate the effect of the independent variable (physical activity) on the dependent variable (depression) when the moderating variable (social support) was not taken into account. As shown in Table 7, the independent variable (physical activity) showed significance (t = −18.866, *p* = 0.000 < 0.05), implying that physical activity has a significant effect on depression, further supporting hypothesis H1. The interaction between the effect of physical activity and social support on mental depression (M2 and M3) revealed a significant change in the value of F from M2 to M3 (F (7966) = 52.174 → F (8965) = 60.008). Additionally, model 3’s interaction term between physical activity and social support was significant (t = −9.144, *p* = 0.000 < 0.05), implying that the magnitude of the effect of physical activity on psychological depression varied significantly depending on the level of the moderating variable (social support).

The slope plot depicts the magnitude (slope) difference in the effect of the independent variable (physical activity) on the dependent variable (psychological depression) when the moderating variable (social support) is varied. The test results show that the specific moderating effect is the difference in the magnitude (slope) of the effect of the independent variable (physical activity) on the dependent variable (psychological depression) when the moderating variable (social support) is taken at different levels. The test results revealed that social support had a stronger effect on mental depression in the high-social-support condition and a weaker effect in the low-social-support condition. This suggests that the predictive effect of physical activity on mental depression in older adults gradually increases with increasing social support. Thus, hypothesis H3 holds true.

## 4. Discussion

### 4.1. Physical Activity and Psychological Depression in Older Adults

This study investigated the relationship between physical activity, self-efficacy, social support, and mental depression among older adults during pandemic prevention and control and the mediating effect of self-efficacy and the moderating effect of social support on the relationship between physical activity and mental depression among older adults. Based on the analysis of the sample situation, it was discovered that the majority of older adults had low levels of exercise and a high depression mood during the pandemic closure period. It was also discovered that there was a negative correlation between physical activity and psychological depression, as evidenced by the correlation analysis between physical activity and psychological depression. That is, the higher the level of physical activity in older adults, the lower the likelihood of them experiencing psychological depression. This is consistent with previous research findings [50,51,52,53]. Physical activity has been shown to produce antidepressant effects via multiple biological and psycho-social pathways primarily through exercise and that physical activity causes changes in the brain to create an environment that prevents depression [54]. Physical activity increases the body’s metabolism, which is beneficial for the production of positive emotions, and it can also diminish unpleasant emotions. Increased social prevention and control, particularly during the COVID-19 pandemic, led to increased rates of depression in older people [55], increasing the need for interventions such as physical activity to regulate adverse moods and alleviate and prevent depressive symptoms. A cross-sectional study of 200 older adults of both sexes found that physical activity increased physical vitality and quality of life, reducing depressive symptoms in older adults [56]. In an analysis of factors influencing depression in older adults in Korea, Kim and Park et al. found that cognitive decline was strongly associated with the development of depression in older adults, and by participating in physical activity, they were able to maintain a healthy lifestyle and reduce the risk of cognitive decline, thereby preventing and reducing depression and improving health in older adults [57]. Zhang and Xiang et al. discovered that physical activity may benefit older adults with depression through both physiological and psychological pathways in a study that reviewed the relationship between older adults and depression. Moderate-intensity exercise can boost metabolism and help people release negative emotions, resulting in more positive mental energy. It can also boost self-esteem and self-efficacy in older people, effectively preventing and treating depression on both the physiological and psychological levels [58]. Thus, physical activity’s positive effects can further influence mental depression in older adults, confirming the research hypothesis that physical activity has a significant negative effect on mental depression in older adults.

### 4.2. Mediating Effect of Self-Efficacy

The study’s findings revealed that during the pandemic prevention and control, the majority of older persons had higher self-efficacy and that self-efficacy mediates the relationship between physical activity and mental depression in older adults, confirming research hypothesis 2. That is, physical activity can indirectly affect mental depression in older adults via self-efficacy, which is consistent with previous findings [59,60], indicating that self-efficacy is a mediator of the effect of physical activity on mental depression. Previous research has found that changes in physical activity cause changes in self-efficacy and that changes in self-efficacy may indirectly mediate the link between physical activity and depression, acting as a potential indirect mediator between physical activity and depression [61]. By investigating exercise training for the treatment of depression in older adults, Barbour and Blumenthal et al. discovered that exercise can serve as an effective treatment for depression in older adults; i.e., it can reduce depression through its effects on self-esteem and self-efficacy in older adults [62]. Similarly, Singh and Clements et al. argued that exercise produces positive emotions in older adults and can increase the indirect psychological benefits of self-efficacy on depression production in older adults [63]. As a result, self-efficacy serves as a mediator in the mechanisms by which physical activity affects depression in older adults. In the present study, older adults affected by pandemic prevention and control were more likely to suffer from depression, particularly those with suspected COVID-19 symptoms [64]. Because of the restrictions on outdoor activities during this time, the elderly’s physical activity was naturally reduced. Participants who were physically active during the lockdown had higher levels of resilience (including an assessment of self-efficacy) and positive attitudes and lower levels of depression, according to Zach and Fernandez-Rio et al. [65]. This study adds to the argument that physical activity levels in older adults during COVID-19 prevention and control affect self-efficacy, which in turn affects psychological depression in older adults, and that self-efficacy plays a mediating role in the relationship between physical activity and psychological depression.

### 4.3. Moderating Effect of Social Support

This study also sought to determine the moderating role of social support in the relationship between physical activity and mental depression in older adults; i.e., when social support is higher in older adults, the effect of physical activity on mental depression is strengthened. In other words, social support can strengthen and even reinforce the relationship between the effects of physical activity and mental depression. This is consistent with previous research findings [66]. It has been demonstrated that the decreased level of physical activity and increased sedentary time of older adults during the COVID-19 closure had a greater negative impact on their mood, resulting in an increased incidence of depression [67], whereas positive changes in social support helped to mitigate the negative impact of the pandemic closure on the mental health of older adults [68]. Possible explanations include the following: On the one hand, social support, as a social determinant of health, can improve physical activity and enhance life satisfaction and subjective well-being in older adults, and various sources of social support (such as friends, family, and neighbors) can have a significant impact on the level of physical activity in older adults [69]; on the other hand, social support can improve the health and well-being of older adults by enabling them to handle stressful situations more effectively and lowering their risk of developing depression [70]. This finding supports the idea that social support has a moderating impact on depressed symptoms in older adults.

### 4.4. Contributions

This research makes three major contributions. First, this study investigated the relationship between physical activity and mental depression in older adults during COVID-19, which broadens the research and theoretical knowledge concerning the study of physical activity and mental depression in older adults. Second, this study investigated the mechanisms through which physical activity acts on mental depression in older adults, demonstrating that self-efficacy mediates between the two and social support moderates between the two. Finally, this study provides a guiding direction and reference for the improvement and prevention of depressive symptoms in older adults through physical activity.

### 4.5. Limitations

This study made various positive contributions to the mental health of older people; however, there are a number of limitations of which it is important to be aware. Firstly, the dimensions of physical activity investigated in this study require expansion, different forms of physical activity were not adequately analyzed in terms of their effects on mental depression in older adults, and the various dimensions of social support were not sufficiently analyzed in terms of their effects on mental depression in older adults. Secondly, as in other cross-sectional analytic studies, this one employed cross-sectional research, which may make it harder to discern the causal association between variables.

Finally, even though the assessments used in the study have been validated, there are still some limitations, especially when it comes to physical activity, which is subjective, because of sample recruitment in a specific context. The use of successfully validated assessments, however, increases the study’s scientific validity and has some implications. The relationship between physical activity and mental depression in older adults should be investigated further in the future through subsequent research designs or pilot studies and increased measures of psychological attributes. In addition to self-mediating efficacy’s role in the relationship between physical activity and mental depression in older adults, additional mediating and moderating variables should be investigated in the future.

## 5. Conclusions

Despite certain limitations, the study found that physical activity in older adults during the prevention and control of the COVID-19 has a significant negative effect on mental depression and plays a key role in the prevention of depression in older adults. After further investigation into the connection between self-efficacy, physical activity, and mental depression in older people, it was discovered that self-efficacy acted as a mediator in this association. Finally, the findings concerning the moderating effect demonstrate that social support could help to moderate the relationship between physical activity and mental depression in older adults. This study adds to the body of evidence elucidating the link between physical activity and mental depression in older adults, with a significant impact on preventing depression in elderly people.

This study contends that self-efficacy’s effect on mental depression in older adults should be highlighted and that long-term physical activity can help prevent and treat depression. It is also critical to emphasize the value of social support in assisting older adults in developing social relationships in order to better cope with negative events and reduce the likelihood of depression.

## Figures and Tables

**Figure 1 ijerph-20-03225-f001:**
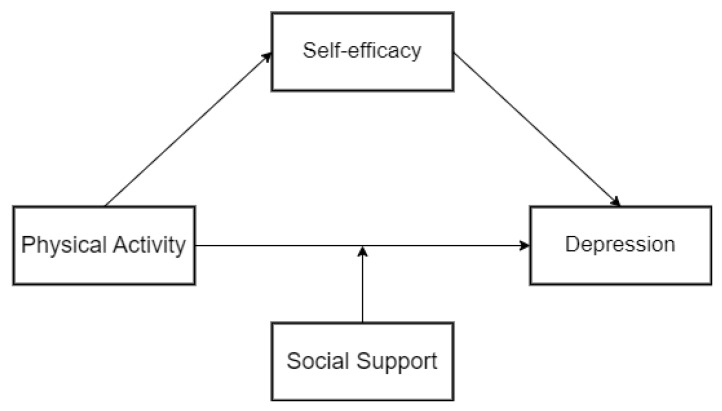
Hypothetical model of the relationship between physical activity and mental depression.

**Figure 2 ijerph-20-03225-f002:**
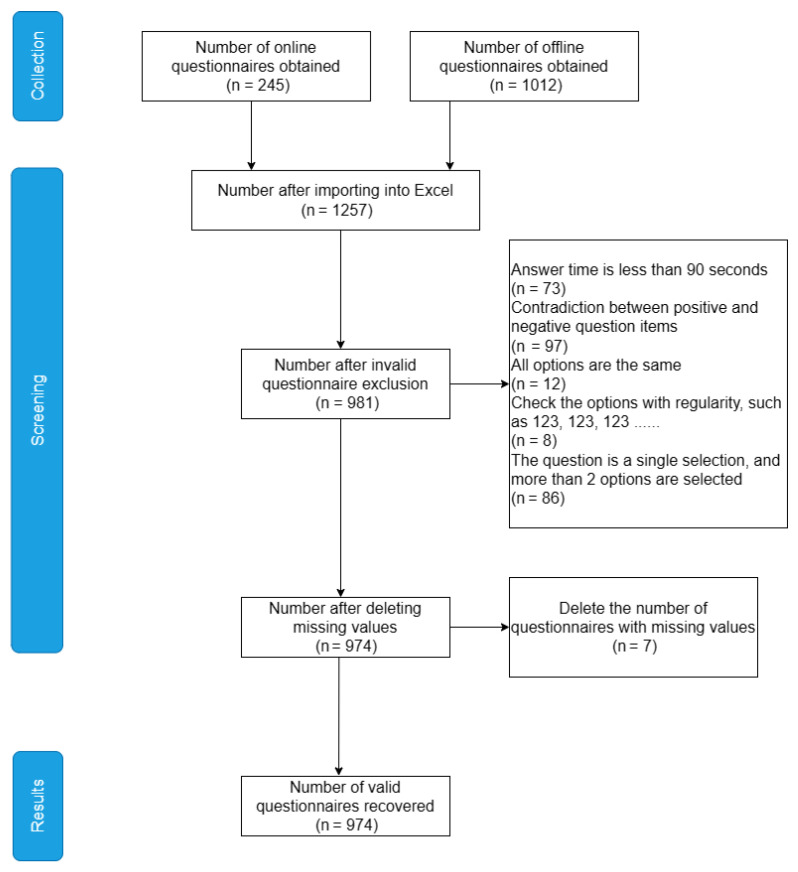
Illustrates the steps for the screening process of the study sample.

**Figure 3 ijerph-20-03225-f003:**
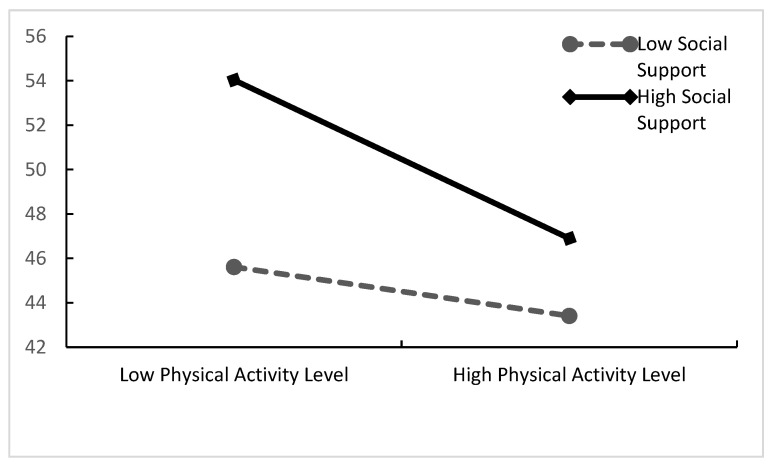
The moderating role of social support between physical activity and mental depression.

**Figure 4 ijerph-20-03225-f004:**
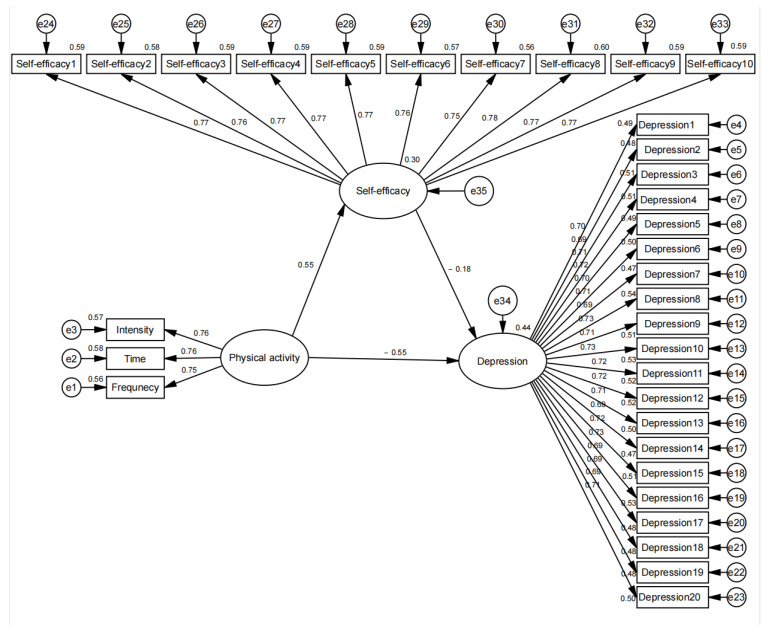
A structural equation model on the role of self-efficacy in mediating between physical activity and mental depression.

**Table 1 ijerph-20-03225-t001:** Demographic characteristics of the sample.

Variable	Frequency	Percentage (%)
Gender		
Male	490	50.308
Female	484	49.692
Age		
60–64	303	31.109
65–69	274	28.131
70–74	229	23.511
75–79	94	9.651
≥80	74	7.598
Education Level		
No schooling	174	17.864
Primary school	261	26.797
Middle school	206	21.15
High school or technical secondary school	167	17.146
College (including higher vocational education)	72	7.392
Bachelor degree	50	5.133
Graduate and higher	44	4.517
Income (USD)		
<2000	253	25.975
2000–3000	292	29.979
3000–4000	156	16.016
4000–5000	175	17.967
>5000	98	10.062
Health condition		
Very Good	438	44.969
Good	146	14.99
Poor	341	35.01
Bad	49	5.031
Physical Activity		
Low exercise levels	422	43.326
Moderate exercise levels	302	31.006
High exercise levels	250	25.667
Mental Depression		
No depression	394	40.452
Suspected depression	66	6.776
Some degree of depression	514	52.772
Self-Efficacy		
Low self-efficacy	374	38.398
Moderate self-efficacy	346	35.524
High self-efficacy	254	26.078
Social Support		
Low social support	152	15.606
Moderate social support	635	65.195
High social support	187	19.199

**Table 2 ijerph-20-03225-t002:** Confirmatory factor analysis.

Dimension	Items	Parameters of Significant Test				SMC	CR	AVE
		Estimate	S.E.	C.R.	*p*-Value			
PA	Frequency	0.746	0.019	39.263	***	0.557	0.798	0.568
	Time	0.764	0.019	40.211	***	0.584		
	Intensity	0.750	0.018	41.667	***	0.562		
Depression	Depression1	0.700	0.015	46.667	***	0.490	0.952	0.501
	Depression2	0.693	0.015	46.200	***	0.480		
	Depression3	0.713	0.015	47.533	***	0.508		
	Depression4	0.716	0.015	47.733	***	0.512		
	Depression5	0.700	0.014	50.000	***	0.489		
	Depression6	0.707	0.014	50.500	***	0.500		
	Depression7	0.688	0.015	45.867	***	0.473		
	Depression8	0.732	0.014	52.286	***	0.535		
	Depression9	0.714	0.014	51.000	***	0.510		
	Depression10	0.726	0.014	51.857	***	0.527		
	Depression 11	0.718	0.014	51.286	***	0.515		
	Depression 12	0.720	0.015	48.000	***	0.518		
	Depression 13	0.705	0.016	44.063	***	0.497		
	Depression 14	0.689	0.016	43.063	***	0.474		
	Depression 15	0.715	0.014	51.071	***	0.511		
	Depression 16	0.729	0.013	56.077	***	0.532		
	Depression 17	0.689	0.014	49.214	***	0.475		
	Depression 18	0.695	0.015	46.333	***	0.483		
	Depression 19	0.693	0.015	46.200	***	0.480		
	Depression 20	0.707	0.015	47.133	***	0.500		
SE	SE1	0.769	0.012	64.083	***	0.591	0.934	0.585
	SE2	0.761	0.013	58.538	***	0.579		
	SE3	0.768	0.012	64.000	***	0.589		
	SE4	0.767	0.013	59.000	***	0.588		
	SE5	0.770	0.012	64.167	***	0.593		
	SE6	0.756	0.013	58.154	***	0.571		
	SE7	0.750	0.014	53.571	***	0.562		
	SE8	0.776	0.012	64.667	***	0.603		
	SE9	0.765	0.013	58.846	***	0.586		
	SE10	0.765	0.012	63.750	***	0.585		
SS	US	0.859	0.016	53.688	***	0.738	0.843	0.646
	SUS	0.882	0.019	46.421	***	0.779		
	OS	0.649	0.022	29.500	***	0.421		

Note: *** *p* < 0.001; SMC, squared multiple correlation coefficient; CR, combination reliability; AVE, average variance extraction; PA, physical activity; SE, self-efficacy; SS, social support; SUS, subjective support; US, utilization of support; OS, objective support.

**Table 3 ijerph-20-03225-t003:** Questionnaire model fitting indicators.

	χ^2^	*df*	χ^2^/*df*	*P*	GFI	AGFI	CFI	NFI	IFI	RMSEA
Model	730.005	588	1.242	0.000	0.960	0.955	0.993	0.964	0.993	0.016

Note: GFI, goodness-of-fit index; AGFI, adjusted goodness-of-fit index; CFI, comparative fit index; NFI, normed fit index; IFI, increased fit index; RMSEA, root mean square error of approximation.

**Table 4 ijerph-20-03225-t004:** Descriptive statistics and correlation of variables.

Variable	M	S.D.	PA	Depression	SE	SS
PA	29.24	22.702	0.753			
Depression	42.98	13.156	−0.649 ***	0.708		
SE	24.57	7.392	0.551 ***	−0.485 ***	0.765	
SS	34.06	10.376	0.233 ***	−0.081 ***	0.199 ***	0.804

Note: *** *p* < 0.001; M, mean; S.D., standard deviation; PA, physical activity; SE, self-efficacy; SS, social support.

**Table 5 ijerph-20-03225-t005:** Mediated effect model fit indices for physical activity, self-efficacy, and mental depression.

	χ^2^	*df*	χ^2^/*df*	*P*	GFI	AGFI	CFI	NFI	IFI	RMSEA
Model	571.176	492	1.161	0.008	0.966	0.961	0.998	0.969	0.996	0.013

Note: GFI, goodness-of-fit index; AGFI, adjusted goodness-of-fit index; CFI, comparative fit index; NFI, formed fit index; IFI, increased fit index; RMSEA, root mean square error of approximation.

**Table 6 ijerph-20-03225-t006:** Mediating effect analysis.

	Path	Estimate	S.E.	Z	Bootstrapping			
					Bias-Corrected		Percentile	
					Lower	Upper	Lower	Upper
Indirect effect	PA → SE → Depression	−0.101	0.023	−4.391	−0.149	−0.058	−0.147	−0.057
Direct effect	PA → Depression	−0.548	0.046	−11.913	−0.638	−0.461	−0.638	−0.461
Total effect	PA → Depression	−0.649	0.034	−19.088	−0.716	−0.583	−0.714	−0.581

Note: Bootstrap sample size is set to 2000. PA, physical activity; SE, self-efficacy.

**Table 7 ijerph-20-03225-t007:** Moderating effect analysis.

	Model 1		Model 2		Model 3	
	*β*	T	*β*	T	*β*	T
Constants	-	24.306 **	-	24.289 **	-	25.725 **
Gender	0.006	0.23	0.009	0.318	0.015	0.552
Age	−0.014	−0.494	−0.015	−0.563	−0.02	−0.775
Education	0.016	0.585	0.017	0.618	0.016	0.604
Income	−0.027	−0.965	−0.027	−0.981	−0.04	−1.515
Health condition	−0.013	−0.464	−0.011	−0.39	0.001	0.034
PA	−0.519	−18.866 **	−0.537	−18.803 **	−0.449	−15.442 **
SS			0.067	2.331 *	0.074	2.618 **
PA×SS					−0.258	−9.144 **
R^2^		0.27		0.274		0.332
Adjusted R^2^		0.266		0.269		0.327
F-value		F (6967) = 56.690, *p* = 0.000		F (7966) = 52.174, *p* = 0.000		F (8965) = 60.008, *p* = 0.000
ΔR^2^		0.27		0.004		0.058
ΔF value		F (6967) = 56.690, *p* = 0.000		F (1966) = 5.435, *p* = 0.020		F (1965) = 83.613, *p* = 0.000

Note: Dependent variable mental depression. * *p* < 0.05 ** *p* < 0.01; PA, physical activity; SE, self-efficacy; SS, social support.

## Data Availability

The data presented in this study are available upon request from the corresponding author. The data are not publicly available due to an ethical agreement with the Chengdu Sport University Social Sciences Ethics Panel to keep them under Ma Xiujie’s personal OneDrive account, which is not accessible to the public.

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
