# Peer review of "The Relationship between Physical Activity and Mental Depression in Older Adults during the Prevention and Control of COVID-19: A Mixed Model with Mediating and Moderating Effects"

_ijerph, 2023, doi:10.3390/ijerph20043225_

Round 1
Reviewer 1 Report
General comment: authors present results of a cross-sectional study to look into the complex effects of physical activity participation on mental health in older adults during COVID-19 prevention and control, and the relationship between physical activity and mental depression in older adults through the mediating effect of self-efficacy and the moderating effect of social support. The study topic is interesting for readers and research. However a number of methodological questions should be responded to or explicitly stated as limitations. A main concern is the unclear and not well described sample and recruitment process and the consistent mix of methodological information and result presentation
Abstract: Authors should not state an effect when associations are documented: “The study's findings revealed that physical activity had a significant negative effect on mental depression in older adults (r = -0.649, p<0.01); Authors request in conclusion a promotion of PA: Physical activity is encouraged for older adults during COVID-19 prevention and control to prevent and alleviate psychological depression. not sure if the associative data may fully back up such requests which may better be evidence based by intervention trials. I suggest to be more careful and more narrowly stick to the data analysed and presented, indicating an association of PA-depressive symptoms (not diagnosed depression) but also social interaction or a mediating association to self-efficacy.
Introduction: In introduction authors well summarize previous evidence on associations between the main variables of the study, describe the medical/social context of the background for the research planned and develop a sound rationale for the study hypothesis. Authors should not state results of the study in introduction: “This study adds to the evidence that social support has both preventive and moderating effects on depression in older adults”. Authors present existing evidence for associations between targeted variables, it is unclear whether they report also mediating effects of variables or mere associations (not stated, presumably not). It may help to further stress previous results on associations analysed by mediation /moderating effects as the method used in the present paper. The objective as stated by authors deviate between abstract and Introduction. I suggest to harmonize.
Methods: Authors analyse a convenient sample with a restricted description of the sample and inclusion criteria ( named “The inclusion standards): (1) people over the age of 60; (2) permanent residents with Chengdu household registration; (3) informed consent and voluntary participation; and (4) unrestricted communication. The exclusion criteria were as follows: (1) a questionnaire completion time of less than 90 seconds; and (2) answers that were identical, duplicate, or invalid. In table 1 they give additional information on age, gender and educational level. By the description given it is hard to understand how participants were recruited and whether the sample was highly selected/ probably not very representative. Not sure if I can follow the criteria for questionnaire completion. Why would a completion time of less than 90 sec be an exclusion ? by which criteria? What would be classified as “identical, dublicate or invalid?” Please comment.
Authors should not mix description of methods and results (table 1). I suggest to put table 1 in results. In table 1 they give additional information on age, gender and educational level. As the sample seems not very well described, authors should at least present baseline data of their variables analyzed including physical activity (PA), status of depression, social status and self efficacy to allow readers to understand the effect of the sample. (e.g. how many would be classified as depressed, less active or have low self efficacy; or none /less of those (ceiling effects).
Authors correctly made a point in introduction that the covid situation represented a most relevant background to the present study (e.g.restricted social interaction especially in China). If available also duration of isolation during the pandemic or other relevant issues concerning the specific pandemic issues may be addressed in the data analysis or at least the sample description (long covid, access to public health services , degree of restricted social interaction etc.). The complexinterqction of Depression and PA may need other established variables for participation on PA, social status or depression (e.g. economic status, social status, health status etc.) to be included if available. The restriction to the 4 variables chosen in the analytic model may not allow a comprehensive evaluation of the data/mechanisms.
The restricted information on inclusion criteria and description of the convenient sample and the limited number of potential influencing factors may represent a limitation of the study. The Physical Activity Rating Scale (PARS-3) revised by Liang, D.C. et al. was used to assess physical activity in the elderly. “The amount of exercise was examined in terms of three aspects: intensity, time, and frequency of participation in physical activity, using a 5-point scale [60]. Specifically, amount of exercise = intensity × time × frequency. Intensity and frequency were scored from 1 to 5 levels recorded as 1 to 5 points, time was scored from 1 to 5 levels recorded as 0 to 4 points, with a maximum score of 100 points and a minimum score of 0 points. The activity level assessment criteria are as follows: 19 for low exercise levels, 20-42 for moderate exercise levels, and 43 for high exercise levels.: Were those cut-offs taken from the original paper?
The description lacks the time span for which PA was assessed. Would different activities be summed up? Were low intensity, short bout habitual activities ,which cover much of older persons PA, included? Representing a major outcome variable of the study the description should be more detailed. As with other PA questionnaires subjective rating of PA is prone to reporting problems. Please comment.
For some variables cut offs for subgroups are suggested (e.G. PA, depression) for some it remains unclear (lickert scaling mentioned but no subgrouping, for others such as social support no subgrouping is mentioned in methods). Later on such subgroups have been used in data analysis(e.g. fig 3, subgroups have been used but no information is given for classification of subgroups. Pa groups are dichotomized and not separated in 3 groups as mentioned in methods). It remains somehow unclear if continuous data or subgroup anaylsis has been used in other analyses.
Results : “Authors describe methodological aspect of the data analysis in results : “(First, validated factor analysis (CFA) was used in this study to test for convergent validity. As shown in Table 2, all question items had factor loadings that ranged from 0.664 to 0.910, all of which were higher than 0.5. Second, to further examine the scale's reliability and validity, average variance extracted values (AVE) and combined reliability (CR) were used. “ I recommend to strictly separate sections (introduction, methods, etc.). see also other comments for this issue Not sure if I support the terminology for reliability and validity. Reliability as stated is not specified and would be “internal reliability “ in my understanding (in contrast to re-test or inter observer reliability) . Construct validity has been used as a term in other analysis testing for association to other variables/domains related to the construct/targeted variable (e.g.motor capacity vs. PA). last sentence is: The questionnaire survey in this research, therefore, has a high level of validity and reliability. Given that there are different aspects of the terms “reliability and validity” the statement is too generalizing. I suggest to introduce the term “internal” for reliability and validity.
Authors state:” It was discovered that each factor's AVE and CR values were greater than 0.5 and greater than 0.7, respectively, indicating that the data had good construct validity and consistency”. Authors did not mention those cu-offs in methods, see comments above / below General comment. Authors used successfully validated assessment methods as a strength of the study. The common understanding would be that those methods would not need to be validated again in each study (there are exception but I would not indicate them here). The question is why authors take the effort to validate measures again and why they would focus specifically on internal reliability and validity and not other relevant biometrical quality standards (e.g. re-test reliability, feasibility, construct validity, ceiling/floor effects etc.). Please comment.
“Table 4 displays descriptive statistics for the main core variables, including means, standard deviations, and correlation coefficients (physical activity, mental depression, self-efficacy, and social support)”. I suggest to put such information in notes below tables.
It would support the understanding to put text directly to associated tables. Table 4 seems a good example that tables have limitations.
Not all abbreviations have been explained in notes (only M, SD) the headings are un clear , partly abbreviations , partly full term (PA Depression SE SS). It seems by the related text, that under such headings correlation coefficients were given (not stated). In my understanding a corresponding table note would briefly describe what is presented and provide all abbreviations for full terms. I assume that values marked by stars would indicate significant results. (not explained). Some tables are extremely “frugal” (Table 5) and are not sufficient to understand analysis without reading the corresponding text Similar remarks relate also to other tables and figures.
In methods a description is lacking which kind of correlation analysis has been used (e.g. spearman or pearson?). In case those established methods were used the values would not correspond to results , indicating rather low values to be a “strong correlation”. “Furthermore, with correlation coefficients of 0.233 and -0.081, respectively, social support had strong positive and negative correlations with physical activity and mental depression, both at a significant level of 0.01 or less”. Such low Coefficients would be considered low to marginal by established cut offs for classification of associations Not sure if I understand the correlations in the matrix table 4 correlating identical! variables with correlation coefficients between between 0.703 and 0.804 indicating similar but not identical data Again in all of chapter “3.4. Moderating Analysis” authors describe relevant methodological issues in results rather than in methods, please separate and match to respective parts of the paper.
Discussion: The discussion takes up present results and for most parts put those in an adequate context and discussion. The reference to the pandemic is relevant but less well supported by the present data (see comments in methods for lack of specific data on pandemic specific data included).
Authors may be more careful when reporting on “psychological depression” participants were not diagnosed but instead results of depressive symptoms were documented. As presumably only a limited number of participants would qualify for clinical depression (see comments on inclusion criteria above) generalizing comments would be not adequate. “This finding supports the idea that social support has a moderating impact on psychological depression in older adults”.
Contributions: Not sure if I support the statement: Finally, this study provides a reference for the improvement and pre-vention of mental depression in older adults through physical activity, so that effective methods can be used to prevent and alleviate mental depression in older adults. Authors present cross-sectional observational data , so results of data analysis on association are presented. This is different from intervention trails providing evidence for effects of an intervention or even a follow up of the course of e.g. the psychological status. (see also authors comments in limitation)
Limitations: Authors correctly state 3 issues as limitation of the study: undetailed information on PA as assessed in the study, the cross-sectional design, and the lack of comprehensiveness in including other mediating/moderation factors in the complex association of PA and depression. Following my comments above I would add the unclear sample and recruitment and the lack of pandemic specific outcomes/variables in this study performed during this specific situation. Assessments used in the study have been validated but still have other limitations specifically the PA assessment (see comments in methods)
Conclusions: I suggest to scale down the statement: the study…. plays a key role in the prevention of depression in older adults. It will still have merits by the data given. Tables (see also comments for tables in text)table figures should have a not include a brief description for the content and a comprehensive list of abbreviations and markers Abbreviations given in note of table 2 do not cover abbreviations given in table and seem disconnected to table content Fig 3: the figure lacks also an explanation for the axis and scores given. In methods cut offs for 3 stages of Pa were given, here the Pa levels are dichotomized, as well as for social support, no cut off for low vs. high social support is given (see also various comments in methods )
Author Response
Dear Reviewer 1:
Thank you for your kind comments on our article, which were very precise and helpful for us. In accordance with your suggestions, we have made a substantial revision of this article. Below, you will find a breakdown of the responses to your comments (in italics):
ABSTRACT:
Comment 1: Authors should not state an effect when associations are documented: “The study's findings revealed that physical activity had a significant negative effect on mental depression in older adults (r = -0.649, p<0.01).
Response:Thank you for your suggestion, We have deleted this statement .(Lines 21-22).
Comment 2: Not sure if the associative data may fully back up such requests which may better be evidence based by intervention trials. I suggest to be more careful and more narrowly stick to the data analyzed and presented, indicating an association of PA-depressive symptoms (not diagnosed depression) but also social interaction or a mediating association to self-efficacy.
Response: Thank you very much for your valuable advice, we have modified this section (Lines 25-27).
INTRODUCTION:
Comment 3: Authors should not state results of the study in introduction: “This study adds to the evidence that social support has both preventive and moderating effects on depression in older adults”.
Response: Thank you very much for your valuable advice, we have redacted this section (Lines 123-124).
METHODS
Comment 4: By the description given it is hard to understand how participants were recruited and whether the sample was highly selected/ probably not very representative.
Response: Thank you very much for your suggestion. We have added some details of the participant recruitment process as appropriate and added a flow chart of the sample screening. (Lines 153-176)
Comment 5: Why would a completion time of less than 90 sec be an exclusion? by which criteria? What would be classified as “identical, duplicate or invalid?”
Response: Your questions is much appreciated. After multiple researchers assessed the time it took to finish the questionnaire, it was discovered that it took more than 90 seconds, so a criterion was established that the questionnaires that were completed in less than 90 seconds would be discarded. The following are the answers to the questions about duplicate and invalid replies: The same answer indicates that the options checked throughout the questionnaire are the same; invalid questionnaires include : (1) scales with counter-questions in which the positive and negative questions are contradictory; and (2) questionnaires with regularity in the options checked, such as 1, 2, 3; 1, 2, 3; 1, 2, 3...... (3) The questions are single choice, yet more than two options are chosen.
Comment 6: Authors should not mix description of methods and results (table 1). I suggest to put table 1 in results. As the sample seems not very well described, authors should at least present baseline data of their variables analyzed including physical activity (PA), status of depression, social status and self-efficacy to allow readers to understand the effect of the sample.
Response: We appreciate your suggestion. We have added Table 1 to the results section and provided the base for the different levels of the four variables of physical activity, psychological depression, self-efficacy and social support of the sample (Lines 267-293).
Comment 7: If available also duration of isolation during the pandemic or other relevant issues concerning the specific pandemic issues may be addressed in the data analysis or at least the sample description (long covid, access to public health services, degree of restricted social interaction etc.).
Response: we appreciate your suggestion. The isolation criteria and minimal period of isolation in medium- and high-risk areas during a pandemic have been included in the paper (Lines 274-282).
Comment 8: The complex interaction of Depression and PA may need other established variables for participation on PA, social status or depression (e.g. economic status, social status, health status etc.) to be included if available.
Response: Thank you very much for your suggestion. We have added income and health condition from the questionnaire to the control variables and have modified the analysis of moderating effects. (Lines 178-181) (Lines 360-382).
Comment 9: The description lacks the time span for which PA was assessed. Would different activities be summed up? Were low intensity, short bout habitual activities, which cover much of older persons PA, included? Representing a major outcome variable of the study the description should be more detailed. As with other PA questionnaires subjective rating of PA is prone to reporting problems. Please comment.
Response: Thank you very much for your suggestion. We have described physical activity in detail, adding in that section the time span of physical activity, as well as the division of intensity and frequency. (Lines 187-195). In this case, the intensity division of exercise includes most of the physical activities of older adults (Lines 187-193). In addition, our study did not summarize the different physical activities, which is one of the limitations of this study.
Comment 10: For some variables cut offs for subgroups are suggested (e.G. PA, depression) for some it remains unclear (Lickert scaling mentioned but no subgrouping, for others such as social support no subgrouping is mentioned in methods). Later on such subgroups have been used in data analysis (e.g. fig 3, subgroups have been used but no information is given for classification of subgroups. Pa groups are dichotomized and not separated in 3 groups as mentioned in methods).
Response: Thank you very much for your suggestion. Since the aim of the study was to verify the correlation between physical activity and depression, including the mediating effect of self-efficacy and the moderating effect of social support, both analyses were done considering the variables as a whole, so subgroup analysis was not used. Concerning Figure 3 (now Figure 4), the levels of physical activity and social support in this figure were calculated using the mean and standard deviation rather than the scale.
RESULTS
Comment 11: Authors describe methodological aspect of the data analysis in results: “(First, validated factor analysis (CFA) was used in this study to test for convergent validity. As shown in Table 2, all question items had factor loadings that ranged from 0.664 to 0.910, all of which were higher than 0.5. Second, to further examine the scale's reliability and validity, average variance extracted values (AVE) and combined reliability (CR) were used. “ I recommend to strictly separate sections (introduction, methods, etc.).
Response: Thank you very much for your suggestion. I have moved that section to the methods section (Lines 232-238).
Comment 12: Last sentence is: The questionnaire survey in this research, therefore, has a high level of validity and reliability. Given that there are different aspects of the terms “reliability and validity” the statement is too generalizing. I suggest to introduce the term “internal” for reliability and validity.
Response: Thank you very much for your suggestion. We have added the word "internal"(Lines 301).
Comment 13: Authors state:” It was discovered that each factor's AVE and CR values were greater than 0.5 and greater than 0.7, respectively, indicating that the data had good construct validity and consistency”. Authors did not mention those cu-offs in methods. The common understanding would be that those methods would not need to be validated again in each study (there are exception but I would not indicate them here). The question is why authors take the effort to validate measures again and why they would focus specifically on internal reliability and validity and not other relevant biometrical quality standards (e.g. re-test reliability, feasibility, construct validity, ceiling/floor effects etc.).
Response: Thank you very much for your suggestion. We have briefly added to this in our methodology (Lines 233-238). It should be noted here that the purpose of the two validations is not the same; the factor loadings of each latent variable measurement equation were analyzed to show the correlation between the factors and the analyzed terms, and the absolute values of the standardized loadings were all greater than 0.6 and showed significance, implying a good measurement relationship, while the values of AVE and CR were analyzed for convergent validity, and the AVE values corresponding to the factors were all greater than 0.5, and CR were all higher than 0.7, which means that the data in this analysis had good convergent validity.
Comment 14: “Table 4 displays descriptive statistics for the main core variables, including means, standard deviations, and correlation coefficients (physical activity, mental depression, self-efficacy, and social support)”. I suggest to put such information in notes below tables.
Response: We appreciate your suggestion, we have modified this (Lines 348-349).
Comment 15: Not all abbreviations have been explained in notes (only M, SD) the headings are un clear, partly abbreviations , partly full term (PA Depression SE SS). Some tables are extremely “frugal” (Table 5) and are not sufficient to understand analysis without reading the corresponding text Similar remarks relate also to other tables and figures.
Response: We appreciate your suggestions. We have added abbreviations to the tables that lack full names (Lines 304-306) (Lines 308-310) (Lines 348-349) (Lines 352-354) (Lines 356) (Lines 383-384).
Comment 16: In methods a description is lacking which kind of correlation analysis has been used (e.g. spearman or pearson?)
Response: Thank you very much for your suggestion. We applied Pearson correlation analysis, mainly to verify the positive and negative relationship between social support and physical activity and psychological depression, mainly based on the positive and negative numbers, while the strength of the correlation does not affect the positive and negative correlation between the two.
Comment 17: Not sure if I understand the correlations in the matrix table 4 correlating identical! variables with correlation coefficients between between 0.703 and 0.804 indicating similar but not identical data.
Response: Thank you very much for your question. There was no correlation between the two data, which were the square root of the average variance extracted (AVE) of the variables, and the AVE square root were greater than the absolute value of the correlation coefficient, indicating good discriminant validity. Since the study was conducted to verify the positive and negative correlations between the variables, this was not stated.
Comment 18: Again in all of chapter “3.4. Moderating Analysis” authors describe relevant methodological issues in results rather than in methods, please separate and match to respective parts of the paper.
Response: Thank you very much for your suggestion. We have adjusted this section (Lines 247-265).
DISCUSSION
Comment 19: The reference to the pandemic is relevant but less well supported by the present data (see comments in methods for lack of specific data on pandemic specific data included).
Response: Thank you very much for your suggestion. We have made changes and added isolation requirements during pandemic to the sample situation analysis (Lines 276-284) (Lines 394-398) (Lines 426-428).
Comment 20: Authors may be more careful when reporting on “psychological depression” participants were not diagnosed but instead results of depressive symptoms were documented. “This finding supports the idea that social support has a moderating impact on psychological depression in older adults”.
Response: Thank you very much for your suggestion. we have modified it (Lines 471).
CONTRIBUTIONS
Comment 20: Not sure if I support the statement: Finally, this study provides a reference for the improvement and pre-vention of mental depression in older adults through physical activity, so that effective methods can be used to prevent and alleviate mental depression in older adults.
Response: Thank you very much for your suggestion. we have modified it (Lines 480-482).
LIMITATIONS
Comment 21: Following my comments above I would add the unclear sample and recruitment and the lack of pandemic specific outcomes/variables in this study performed during this specific situation. Assessments used in the study have been validated but still have other limitations specifically the PA assessment (see comments in methods).
Response: we appreciate your suggestion. We have added this limitation to the manuscript (Lines 489-494).
CONCLUSIONS
Comment 22: I suggest to scale down the statement: the study…. plays a key role in the prevention of depression in older adults.
Response: we appreciate your suggestion. We have trimmed this section (Lines 509-516).
Comment 23: Tables (see also comments for tables in text)table figures should have a not include a brief description for the content and a comprehensive list of abbreviations and markers Abbreviations given in note of table 2 do not cover abbreviations given in table and seem disconnected to table content.
Response: Thank you very much for your suggestion. We have added abbreviations to the table that lack full names (Lines 304-306) (Lines 308-310) (Lines 348-349) (Lines 352-354) (Lines 356) (Lines 383-384).
Comment 24: Fig 3: the figure lacks also an explanation for the axis and scores given. In methods cut offs for 3 stages of Pa were given, here the Pa levels are dichotomized, as well as for social support, no cut off for low vs. high social support is given (see also various comments in methods).
Response: Thank you very much for your suggestion. The division between high and low levels of physical activity and social support is described in our methodology (Lines 263-267). The data for the vertical axis is automatically generated during the plotting process and is mainly calculated for the four points at the ends of the two lines, which are calculated during the analysis. And concerning Figure 3 (now Figure 4), the levels of physical activity and social support in this figure were calculated using the mean and standard deviation rather than the scale.
We hope you are pleased with the revised manuscript, which also includes the changes expected from other peer reviewers.
Thanks again, and kind regards.

Reviewer 2 Report
The article is interesting; however, it needs some clarifications.
INTRODUCTION
1. The introduction needs improvement; there is no academic structure of the paragraphs. Three elements are essential here: What is known, what is unknown about the research question, and why the study was needed to carry out. The paragraphs must have an academic structure with an opening sentence, support sentences, and closing statement (sandwich structure).
2. It is important to reformulate the second paragraph "what is not known about the topic" in which the gaps or limitations that exist in the literature about the research question should be described. It is essential to describe the methodological deficiencies in previous studies, for example, unmeasured confounding, uncontrolled confounding, small sample size, lack of statistical power, studies without sufficient biostatistical rigor, among others.
METHODS
1. Authors need to develop a participant selection flowchart. It is necessary to know in detail the number of participants excluded, according to each criterion.
2. The authors should report data used to estimate the study sample. In addition, provide more detail on the type of sampling used.
3. I appreciate the authors for having estimated the internal consistency of the instruments. However, psychometric properties should be added: validity and reliability measures, as reported in similar studies.
4. The analysis plan section is very limited and lacks explanation on what processes were performed for the descriptive, bivariate and simple/multiple regression analysis. In the description, the authors should briefly mention the evaluation of statistical assumptions and the potential evaluation of collinearity between the variables of interest.
DISCUSSION
1. The limitations section should be substantially improved. The authors should mention the epidemiological biases, the potential solution they have given to prevent these biases from invalidating their studies, etc. Authors should examine whether their study has selection bias, are the findings representative for the study population, is there confounding bias due to variables that are potentially associated with their outcome and have not been measured?
2. The authors should add a strengths section immediately following the limitations section.
Author Response
Dear Reviewer 2:
Thank you for your kind comments on our article, which were very precise and helpful for us. Following your suggestions, we have made a substantial revision to this article. Below, you will find a breakdown of the responses to your comments (in italics):
INTRODUCTION:
Comment 1: The introduction needs improvement; there is no academic structure in the paragraphs.
Response: Thank you very much for your suggestion. We have revised the content of the introduction. (Lines 31-72)
Comment 2: It is important to reformulate the second paragraph "what is not known about the topic", in which the gaps or limitations that exist in the literature about the research question should be described.
Response: Thank you very much for your suggestion. We have added the limitations and shortcomings of the relevant literature in the manuscript's introduction. (Lines 53-62)
METHODS
Comment 3: Authors need to develop a participant selection flowchart. It is necessary to know the number of participants excluded, according to each criterion.
Response: Thank you very much for your suggestion. We have added a flow chart for sample screening to the manuscript. (Lines 177-178)
Comment 4: The authors should report data used to estimate the study sample. In addition, provide more detail on the type of sampling used.
Response: Thank you very much for your suggestion. We have added this section and described it in detail (Lines 148-178).
Comment 5: However, psychometric properties should be added: validity and reliability measures, as reported in similar studies.
Response: We appreciate your suggestion. I am sorry to inform you that since this study is not an experimental study, the measurement of psychological attributes is not addressed at this time, which may be used in our future studies. We have added this to our future research directions (Lines 232-268).
Comment 6: The analysis plan section is minimal and lacks an explanation on what processes were performed for the descriptive, bivariate and simple/multiple regression analysis.
Response: Thank you very much for your suggestion. We have corrected this section in our methodology (Lines 232-268).
Comment 7:In the description, the authors should briefly mention the evaluation of statistical assumptions and the potential assessment of collinearity between the variables of interest.
Response: Thank you very much for your suggestion. We have made the appropriate changes in the manuscript (Lines 318-330).
Comment 8: The limitations section should be substantially improved. The authors should mention the epidemiological biases, the potential solution they have given to prevent these biases from invalidating their studies, etc.
Response: We appreciate your suggestion. We have improved on the limitations (Lines 498-501).
Comment 9: The authors should add a strengths section immediately following the limitations section.
Response: Thank you very much for your suggestion. We have added an advantage section after the limitation section (Lines 501-503).
We hope you are pleased with the revised manuscript, including the changes expected from other peer reviewers.
Thanks again, and kind regards.

Round 2
Reviewer 2 Report
The authors have substantially improved the manuscript, based on the major observations noted in the peer review.